# ALCUNA: Large Language Models Meet New Knowledge

**Xunjian Yin**[∗] and **Baizhou Huang**[∗] and **Xiaojun Wan**
Wangxuan Institute of Computer Technology, Peking University
Center for Data Science, Peking University
The MOE Key Laboratory of Computational Linguistics, Peking University
{xjyin,hbz19,wanxiaojun}@pku.edu.cn

## Abstract

With the rapid development of NLP, large-scale language models (LLMs) excel in various tasks across multiple domains now. However, existing benchmarks may not adequately measure these models' capabilities, especially when faced with new knowledge. In this paper, we address the lack of benchmarks to evaluate LLMs' ability to handle new knowledge, an important and challenging aspect in the rapidly evolving world. We propose an approach called **Know-Gen** that generates new knowledge by altering existing entity attributes and relationships, resulting in artificial entities that are distinct from real-world entities. With KnowGen, we introduce a benchmark named **ALCUNA** to assess LLMs' abilities in knowledge understanding, differentiation, and association. We benchmark several LLMs, reveals that their performance in face of new knowledge is not satisfactory, particularly in reasoning between new and internal knowledge. We also explore the impact of entity similarity on the model's understanding of entity knowledge and the influence of contextual entities. We appeal to the need for caution when using LLMs in new scenarios or with new knowledge, and hope that our benchmarks can help drive the development of LLMs in face of new knowledge.

## 1 Introduction

Large-scale language models (LLMs) have made impressive progress in the last few years (Brown et al., 2020; Ouyang et al., 2022; Touvron et al., 2023; OpenAI, 2023), which perform surprisingly well on various tasks on various domains, to the extent that many traditional benchmarks (Thorne et al., 2018; Wang et al., 2018) are no longer sufficient to measure the capabilities of LLMs. Therefore, some new benchmarks have been proposed to evaluate the ability of the model to solve more complex tasks such as college entrance exams, law

---

[∗]These authors contributed equally to this work.

school admission tests, math competitions and so on (Hendrycks et al., 2021; Guo et al., 2023; Zhong et al., 2023). LLMs also achieve promising results on these benchmarks.

However, it is surprising that there is not yet a benchmark to evaluate the ability of large models in face of new knowledge, which is very important and challenging. Why is this evaluation important? Firstly, we are in a changing world, where models encounter new knowledge frequently in practice. And some work (Peng et al., 2023) is exploring retrieval methods to augment large models, which will also cause the models to meet new knowledge frequently. So of course we expect the model to perform well in such situations, because re-training the model every time is very expensive and unrealistic. Secondly, as Elangovan et al. (2021) mentioned, the presence of overlap between the training and test data can lead to a misestimation of the model's memory ability as generalization ability, especially nowadays when LLMs are trained on an enormous amount of data. Whereas, evaluation on new knowledge does not need to worry about such data leakage, as new knowledge can usually lead to new data and thus more reliable and valuable assessment of the model's ability.

While such evaluation is important, it is challenging to construct the benchmark. The reason is that it is difficult to ensure that the knowledge contained in the benchmark is new for LLMs, since training data for some models are large and non-publicly available. Furthermore, it is also difficult to ensure that the knowledge used for benchmarking will be not outdated and inefficient, as there are many LLMs that may soon include data from the benchmark in their training. In summary, such benchmark for new knowledge needs to exhibit three basic characteristics: it contains enough new knowledge for sufficient evaluation (**sufficient**), the knowledge is new to all models (**model-agnostic**) and the knowledge can remain new for a long time

(**long-lasting**).

There are several possible solutions to the above challenges. One option is to always use the most updated data such as the news of the day (temporal knowledge). However, this is both labor-intensive to race against the LLM training and unclear about the lifetime of the proposed data. Another option is to keep the benchmark closed-source, with an authoritative committee managing the data and users calling API when evaluating, thus preventing using them for training LLMs . To reach this point further requires community coordination.

To better address these challenges, we propose an approach to GENerate new KNOWledge conveniently (**KnowGen**) based on the structured representation of existing entity knowledge by making reasonable changes to entity attributes and relationships. There are differences and associations between artificial entities and existing entities. Particularly, we apply KnowGen with structured biological taxonomic information data to rationally create a group of organisms that do not exist in the world. To test the model's ability in face of new knowledge, we construct a variety of questions about these artificial entities that can examine the model's ability to understand new knowledge (**Knowledge Understanding**), distinguish between model's internal knowledge and new knowledge (**Knowledge Differentiation**) and make multi-hop reasoning by linking model's internal and new knowledge (**Knowledge Association**). We use the Artificial**Ly** Constr**U**cted k**N**owledge to **A**ssess LLMs as a benchmark (**ALCUNA**).

We evaluate and analyze several popular large models based on ALCUNA, including ChatGPT[1], Alpaca, Vicuna, and ChatGLM with vanilla, CoT (Chain-of-Thought), Zero-Shot and Few-Shot settings (Brown et al., 2020; Kojima et al., 2023; Wei et al., 2023). We find that neither ChatGPT nor other models perform very well in face of new knowledge. ChatGPT does a good job of understanding and differentiating new knowledge, but almost all models fail to reason between the new knowledge and the internal knowledge. This reminds us to remain cautious when large models encounter new scenarios and knowledge. In addition, we explore the impact of entity similarity on the model's understanding of entity knowledge, the impact of contextual entities, etc.

The contributions of our work are listed below:

1) we propose a new method KnowGen to generate new knowledge for simulating real scenarios. 2) we apply our method to produce an artificial biological entity knowledge dataset, ALCUNA, as a benchmark for evaluating the performance of models in face of new knowledge. 3) we evaluate and analyze several popular large models and obtain insightful observations and conclusions.

Our benchmark has been released to the community to facilitate future research [2].

---

**Algorithm 1:** Knowledge Generation

**input** : One Class $C$
**output** : Property Set $\mathcal{T}(\tilde{e})$ of $\tilde{e}$

$e^p \leftarrow$ RandomSelect$(C)$ ;
$E^{psb} \leftarrow$ sib$(e^p)$
// Get the triplet set for heredity, variation, dropout and extension
$\mathcal{T}_R^h, \mathcal{T}_R^v, \mathcal{T}_R^d \leftarrow$ RandomSplit$(\mathcal{T}_R(e^p))$
$\mathcal{T}_A^h, \mathcal{T}_A^v, \mathcal{T}_A^d \leftarrow$ RandomSplit$(\mathcal{T}_A(e^p))$
$\mathcal{T}^e \leftarrow$ RandomSample$(\mathcal{T}(E^{psb}))$

// **Heredity** and **Dropout**
$\mathcal{T}_R(\tilde{e}) \leftarrow \mathcal{T}_R(C) \cup \mathcal{T}_R^h$ ;
$\mathcal{T}_A(\tilde{e}) \leftarrow \mathcal{T}_A(C) \cup \mathcal{T}_A^h$

// **Variation**: replacing the object with an entity from the same class
**for** $(e^p, r, e')$ *in* $\mathcal{T}_R^v$ **do**
   $E'^{sb} \leftarrow$ sib$(e')$
   $e'^{sb} \leftarrow$ RandomSelect$(E'^{sb})$
   $\mathcal{T}_R(\tilde{e}) \leftarrow \mathcal{T}_R(\tilde{e}) \cup \{(\tilde{e}, r, e'^{sb})\}$

// **Variation**: add gaussian noise to the value or copy from $E^{psb}$
**for** $(e^p, a, v)$ *in* $\mathcal{T}_A^v$ **do**
   **if** isnum$(v)$ **then**
      $\tilde{v} \leftarrow v + \mathcal{N}(0, v/10)$
   **else**
      $e^{psb} \leftarrow$ RandomSelect$(E^{psb})$
      $\tilde{v} \leftarrow$ GetValue$(e^{psb}, a)$
   $\mathcal{T}_A(\tilde{e}) \leftarrow \mathcal{T}_A(\tilde{e}) \cup \{(\tilde{e}, a, \tilde{v})\}$

// **Extension** and get final property
$\mathcal{T}(\tilde{e}) \leftarrow \mathcal{T}_A(\tilde{e}) \cup \mathcal{T}_R(\tilde{e}) \cup \mathcal{T}^e$

---

## 2 KnowGen: Knowledge Generation

In this section, we begin by presenting our inspiration, then formally introduce our knowledge generation method KnowGen.

### 2.1 Inspiration

According to the ontological form (Sowa, 1995; Noy et al., 2001), we can represent most knowledge

---

[1] https://chat.openai.com/chat

[2] https://github.com/Arvid-pku/ALCUNA

as entities, the classes to which they belong, their attributes and the relations between them. And inspired by organisms: organisms can produce organisms with new properties naturally through heredity and variation. *Can knowledge also be "inherited" and "varied" in a broader context?* Different entities of the same class have different properties as well as commonalities. Generally speaking, entities of the same class are similar to some extent, while they have some different properties. By analogy with biological terminology, we refer to this characteristic of knowledge of similar entities as "hybridizability". This inspires us to use different entities of the same class to fuse their properties, simulating the process of biological inheritance and variation, to generate new entity knowledge.

In the following we will formally define and describe how knowledge is "inherited" and "varied" in our approach.

## 2.2 Knowledge Formalization

Based on the design of the ontology, We represent knowledge from the viewpoint of Entity. Entity $e \in \mathcal{E}$ is a distinct and identifiable object or individual in the world. Each entity can possess various attributes $a \in \mathcal{A}$, which describe the properties of the entity with a value $v \in \mathcal{V}$. At the same time, each entity can participate in relations $r \in \mathcal{R}$ with other entities. Both the attributes and relations of entity $e$ can be represented as a set of property triplets: $\{(e, a, v)\} = \mathcal{T}_A(e) \subset \mathcal{T}_A$ and $\{(e, r, e')\} = \mathcal{T}_R(e) \subset \mathcal{T}_R$. Entities with similar characteristics may be aggregated into a class $C \subset \mathcal{E}$. For convenience, we denote the same properties across all entities in class $C$ as $\mathcal{T}(C) = \mathcal{T}_A(C) \cup \mathcal{T}_R(C)$. Without any special description, the triplet $\mathcal{T}(e)$ of entity $e$ refers to its unique properties.

For example, Figure 1 shows an example in the form of such structured knowledge, where both Alpaca and Vicuna are existing entities belonging to the class Camels and Alcuna is an artificial entity generated by us. Alpaca has attributes such as "Body mass" and relations such as "Eaten by", and can be formed into triplets such as (Alpaca, Body mass, 60kg) and (Alpaca, Eaten by, Cougar).

## 2.3 Construction of New Entities

Focusing on knowledge in the form of ontology, we aim to construct artificial entities reasonably to accelerate the process of new knowledge generation in the real world. When creating an artificial entity

within a specific class, it must adhere to certain common properties shared by other entities in the same class. Furthermore, it is essential for the artificial entity to possess some unique properties that differentiate it from existing entities. To address these requirements for individuality and commonality, we propose a fast and effective method to construct an artificial entity that fuses attributes and relations of entities within the same class.

Initially, we select an existing entity $e^p$ from a specific class $C$ to serve as the *parent entity* for our artificial entity $\tilde{e}$ and consider other entities within the same class $C$ as *sibling entities of parent*, denoted by $sib(e^p) = \{e_1, ...e_n\}$. Our goal is to construct an artificial entity that exhibits high similarity to the parent entity and conforms to the commonality of the class (*heredity*) while incorporating properties from the sibling entities or reasonably changing property values (*variation*). As an example, in Figure 1, the artificial entity Alcuna inherits the "Diet" and other attributes from the parent Alpaca, while the "Body mass" is varied.

Besides the above operations of heredity and variation, we construct new entities with additional *extension* and *dropout* of the properties of the new entities, in order to mimic human progressive cognitive processes of entities. As the example in Figure 1 shows, we extend the attribute of "First appearance" from Vicuna to Alcuna, and drop out the "Life span" from the parent entity Alpaca.

The whole method can be seen in Algorithm 1. A detailed description of KnowGen with natural language and expressions is shown in Appendix A. The entities constructed in this way are not only reasonable but also convenient for us to assess the model's cognitive ability to associate and differentiate new knowledge with the existing knowledge.

## 2.4 Question Answering as Evaluation Task

Based on the constructed artificial entities, a natural form of evaluation is to ask questions to LLMs in the context of new knowledge. In specific, we leverage an attribute triplet $(\tilde{e}, a, v)$ for generating a one-hop question $q(\tilde{e}, a, v)$ by specifically asking about the object $v$, given the subject $\tilde{e}$ and attribute $a$. With a chain of relation triplets $\mathcal{T}_C = (\tilde{e}, r, e_1) \rightarrow (e_1, r_1, e_2) \rightarrow ... \rightarrow (e_{N-1}, r_{N-1}, e_N)$, we construct a multi-hop question $q(\mathcal{T}_C)$ asking about the tail object $e_N$, given the head subject $\tilde{e}$ and relations $\{r, r_1, ..., r_{N-1}\}$.

We propose that LLM requires the knowledge understanding, knowledge differentiation and

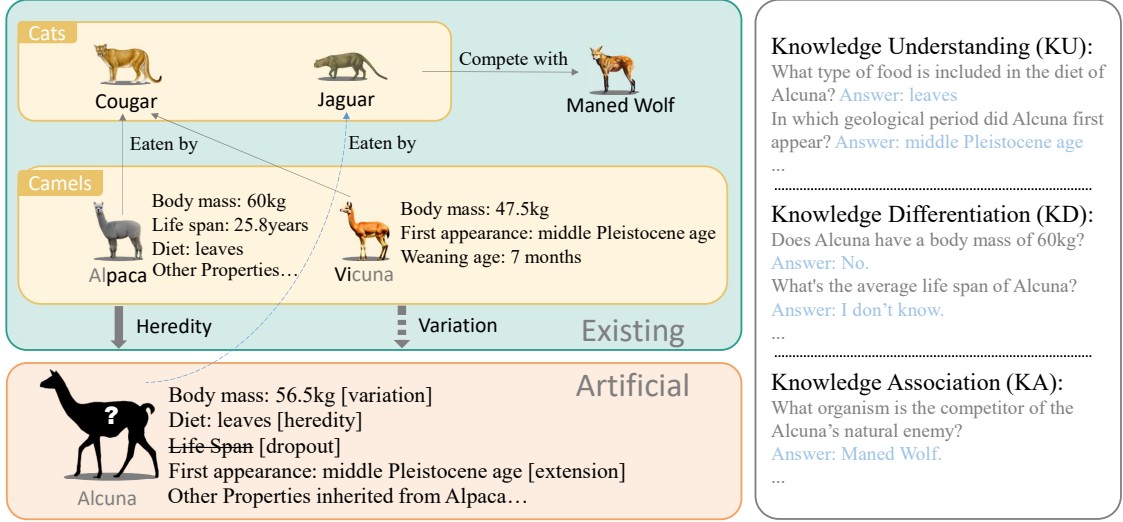

Figure 1: Demonstration of ALCUNA, including heredity, variation, extension and dropout operations in KnowGen, generated artificial entity named Alcuna and three types of questions related to it.

knowledge association abilities when faced with new knowledge. To enable a more detailed assessment, we design diverse categories of questions to evaluate each ability, as shown on the right side of Figure 1.

Specifically, we sample some attribute triplets in the variation set $\mathcal{T}_A^v(\tilde{e})$ and the dropout set $\mathcal{T}_A^d(\tilde{e})$ to create KD question set, which is ideally suited for evaluating the knowledge differentiation ability of a model to distinguish between the parent entity $e^p$ existing in its internal knowledge and the newly constructed artificial entity $\tilde{e}$ provided in the context.

The proficiency of LLMs in reasoning along mastered knowledge graph has been demonstrated in previous studies (Hu et al., 2022; Choudhary and Reddy, 2023). However, it remains uncertain whether it can effectively establish connection between newly encountered artificial entity and the existing knowledge for multi-hop reasoning task. To investigate this, we incorporate the relations of artificial entity to construct a new graph, which encompasses both existing entities and the artificial entity. We then perform a breadth-first-search on the relation graph to identify a chain of relation triplets $\mathcal{T}_C$ with the artificial entity serving as root (e.g., [*(Alcuna, Eaten by, Jaguar), (Jaguar, Compete with, Maned Wolf)*]), and then utilize the chain to generate a multi-hop question $q(\mathcal{T}_C)$ (e.g., *What organism is the competitor of the Alcuna's natural enemy?*). We group such questions into KA question set.

For the rest of the artificial entity's property

triplets, we utilize them to evaluate the ability of remembering and understanding new knowledge by simply asking questions about the objects in triplets. We group such questions into KU question set.

## 3 ALCUNA: Our Benchmark

With our proposed method, one can create a large amount of new entities quickly based on existing structured knowledge. A natural attempt is to construct new organisms on already discovered ones, since the biological taxonomy is perfectly adapted to our proposed approach. Therefore, we utilize KnowGen to propose ALCUNA, a biological dataset for evaluating the ability of the model in face of new knowledge as shown in Figure 1.

### 3.1 EOL Database

We utilize the structured data from the EOL[3] (Encyclopedia of Life) database (Parr et al., 2014) to provide existing knowledge, which is an online, freely accessible database that aims to provide information about all known species on Earth. EOL organizes all biological taxons into a taxonomic tree, in which each entity belongs to a class. The most intriguing feature of EOL is that it constructs rich structured data in the form of key-value pairs for each biological entity, including taxonomic rank, attributes, relationships and information source. As a whole, it contains 2404790 entities with a total of 13625612 properties consisted of 669 property types. The substantial volume of data, coupled with

---

[3] https://eol.org/

its well-organized format, renders EOL the most suitable data source for constructing ALCUNA.

## 3.2 Artificial Entity Construction Details

Each time we select a class $\mathcal{C}$ from the taxonomic tree of EOL and consider its members as entities. We then divide them into one parent entity $e^p$ and its siblings $sib(e^p)$ from which we construct the artificial entity. Since a high-level taxon is usually not a specific organism, the properties it possesses may be too homogeneous, so we screen out all taxons belonging to kingdom, phylum and domain.

In the real world, the naming scheme of a newly found creature usually incorporates the same prefix or suffix of other creatures of the same species. In order to mimic the real world scenario and considering the tokenization algorithms used in LLMs, we firstly split the names of related existing entities (i.e. the parent entity and sibling entities of parent) into subwords [4]. Then we randomly select names of related entities, and for the $i$-th selected entity we choose its $i$-th subword. For example, "ALCUNA" is created from Alpaca and Vicuna.

## 3.3 Template-based Question Generation

Given a property triplet $(\tilde{e}, a, v)$ (one-hop setting) or a chain of property triplets $\mathcal{T}_C = (\tilde{e}, r, e_1) \rightarrow (e_1, r_1, e_2) \rightarrow ... \rightarrow (e_{N-1}, r_{N-1}, e_N)$ (multi-hop setting), we aim to generate natural language question asking about the tail object.

We leverage ChatGPT in the process of question generation to avoid expensive labor costs following Petroni et al. (2019). Specifically, we use ChatGPT to generate a question template with a placeholder [T] given only the relevant properties to avoid introducing too much model's knowledge of a specific entity. Then we generate questions from the question template by replacing [T] with the name of head subject. We generate five question templates for each property group in form of multiple choice, fill-in-the-blank and Boolean questions. The details about the prompts used for question generation and examples are shown in Appendix B.2. To ensure the quality of automatic question generation by this method, we randomly sample 100 questions each for one-hop and multi-hop questions for human checking. It turns out that for the generated one-hop questions, 98% are correct; for the multi-hop questions, 95% are correct. It shows that this way of constructing questions is acceptable.

[4]We utilize the tokenizer of GPT-2 for tokenization.

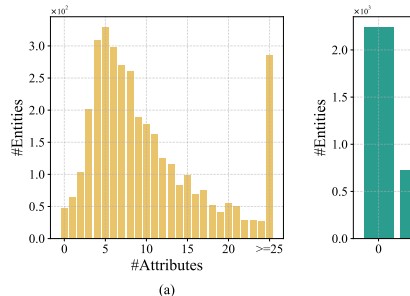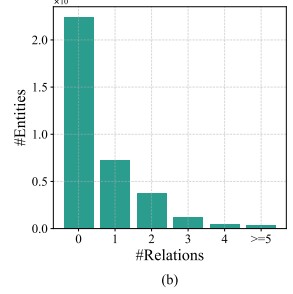

Figure 2: (a) The number of entities with different counts of attributes. (b) The number of entities with different counts of relations.

## 3.4 Dataset Summary

With the previous steps, we constructed a dataset, ALCUNA, for evaluating the ability of LLMs in face of new knowledge. The ALCUNA dataset consists of a total of 84351 questions about 3554 artificial entities. We ensure that the constructed artificial entities contain rich and unique attributes and relationships by filtering out parent entities with less than three properties. Specifically, each artificial entity contains 11.75 property triples and 25.39 siblings on average. The distribution of the number of property triplets is shown in Figure 2.

We organize the dataset in terms of questions, and for each question we collect the corresponding property triplets as evidence and the relevant artificial entities' information as new knowledge. We divide all questions into three subsets, KU, KD and KA, as mentioned in Section 2.4 to measure the corresponding capabilities of LLMs in a fine-grained manner. In specific, KU, KD and KA contain 11316, 27186 and 15353 questions respectively. The details about question forms in the three subsets are shown in Appendix B.1.

## 4 Evaluation of LLMs

### 4.1 LLMs Selected for Evaluation

We select several popular LLMs for evaluation and analysis on our benchmarks, including ChatGPT, Alpaca-7B, Vicuna-13B and ChatGLM-6B. The detailed description of our selected models can be found in Appendix C.

### 4.2 Evaluation Methods

In order to adapt the approach in the era of large models and to match the application scenarios in practice, we introduce two types of evaluation methods: zero-shot and few-shot. We implement both the vanilla and "Chain-of-Thought" (CoT) reasoning forms for zero-shot and few-shot setting.

For experiments in the zero-shot setting, our inputs are structured representations of new knowledge and questions to be answered by the model. For experiments in the few-shot setting, we include several examples of the same form together with the answers, which we hope will help the model understand. For the zero-shot CoT, we append "Let's think step by step." at the end of questions, and for the few-shot CoT, the reasoning process for the answers of examples is also attached. Please refer to Appendix D for the detailed method description. An example of prompt used in our experiment is shown in Table 12.

## 4.3 Evaluation Metric

Since our questions are in the form of multiple choice, fill-in-the-blank or Boolean questions, the golden answers are usually just one or a few words. Therefore, we determine the correctness of the model output by matching it with the answer (like Accuracy). Since there may be more than one possible answer to some questions, such as those asking about the geographical distribution of entities, etc., we consider the answer to be correct as long as it matches one of the correct answers. This is a less stringent measurement, but as seen in Section 5, most models still perform poorly. Using the proposed dataset, we evaluate each model's ability with each method for knowledge understanding, knowledge differentiation, and knowledge association, respectively. We report the average score on the entire benchmark in each setting.

## 4.4 Data Filtering

Since there are differences of internal/existing knowledge of different models due to the different model size and training data, we further filter the samples in our dataset for each model before testing, based on previous work (Petroni et al., 2019), in order not to be influenced by the difference (which is not the theme of our paper) and to *compare the models' performance in face of new knowledge in a more focused and fair way*. For our method of filtering questions, please refer to Appendix E. We experiment and analyze the four models mentioned in Section 4.1 based on the filtered new knowledge, using the evaluation settings introduced in Section 4.2.

## 5 Result and Analysis

### 5.1 Overall Results

The performance of the LLMs on our benchmark under different settings is shown in Table 1. We can see that ChatGPT has the best performance in all settings, which is consistent with our usual beliefs. Vicuna has the second best performance among all models. In terms of methods, the few-shot setting performs better than the zero-shot setting overall, and CoT performs better than the vanilla form in most cases.

In face of new knowledge, as seen in Table 1, LLMs do perform poorly except for ChatGPT on KU and KD experiments. Among all abilities, knowledge association is obviously the most difficult for LLMs, and all of them have difficulty in relating to their internal knowledge through new knowledge provided, and thus in making multi-hop reasoning correctly. The performance of knowledge understanding and knowledge differentiation is better than that of knowledge association, but yet not satisfactory for most LLMs.

In summary, current LLMs perform relatively poorly in face of new knowledge, slightly better in knowledge understanding and knowledge differentiation, and have more difficulty in reasoning across new and existing knowledge. In order to have a clearer view of models output, please refer to Appendix F for the analysis of models output.

Considering that it is expensive, slow and unstable to call ChatGPT's API, without loss of generality, all the following comparison experiments for analysis are conducted on three other models. In addition, for convenience, the following analysis experiments are performed in the setting of the vanilla few-shot method, and structured input artificial entity knowledge, if not specifically stated.

### 5.2 Impact of Entity Similarity

In this section we explore the effect of the similarity between the artificial entity and the parent entity on the model performance over the KD questions, which are designed to assess the model's ability to distinguish between new knowledge and existing knowledge. Specifically, we explore attribute similarity and name similarity.

**The More Similar, the More Confusing (unless powerful enough)** We define the proportion of overlap of properties between entities as the property similarity of entities. As shown in Figure 3,

|  | ChatGPT | Alpaca | Vicuna | ChatGLM | ChatGPT | Alpaca | Vicuna | ChatGLM |
|---|---|---|---|---|---|---|---|---|
|  | Zero-Shot-Vanilla | | | | Zero-Shot-CoT | | | |
| **KU** | 50.19 | 31.02 | 34.12 | 34.64 | 68.75 | 39.81 | 39.61 | 24.85 |
| **KD** | 58.70 | 15.35 | 38.65 | 32.29 | 61.78 | 14.29 | 38.53 | 22.84 |
| **KA** | 28.44 | 24.60 | 29.71 | 10.29 | 35.36 | 19.66 | 29.55 | 4.97 |
| **Avg.** | 52.85 | 25.12 | 35.98 | 31.26 | 63.34 | 29.44 | 38.00 | 22.04 |
|  | Few-Shot-Vanilla | | | | Few-Shot-CoT | | | |
| **KU** | 75.44 | 33.80 | 41.22 | 47.97 | 82.18 | 40.77 | 43.67 | 40.91 |
| **KD** | 64.20 | 38.97 | 46.76 | 41.42 | 74.99 | 36.24 | 55.81 | 37.19 |
| **KA** | 41.52 | 27.63 | 30.10 | 27.47 | 37.88 | 25.73 | 25.07 | 26.93 |
| **Avg.** | 64.37 | 35.09 | 42.74 | 42.56 | 74.11 | 38.02 | 47.73 | 37.64 |

Table 1: Performance of LLMs at our benchmark under different settings

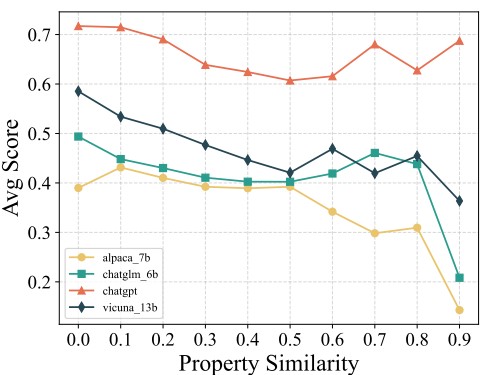

Figure 3: Relationship between model performance on KD questions and the property similarity between the artificial entity and its parental entity.

|  | Alpaca | Vicuna | ChatGLM |
|---|---|---|---|
| similar | 38.74 | 47.23 | 41.61 |
| random | 39.52 | 47.64 | 43.20 |

Table 2: Results on KD questions of the entity with the same property and different name.

we divide questions according to the property similarity between the parental entities and the artificial entities, and calculate the scores of models in different similarity ranges on the KD problem respectively. We can find that the performance of the robust ChatGPT to differentiate entities is almost independent of the similarity. But other than it, all other models are affected by entity similarity. The more similar the new entities are to the existing entities, the harder they are to differentiate them, which illustrates the flaws of LLMs. The further analysis is shown in Appendix **??**

**The Name also Plays a Role** Since each artificial entity is identified by its name, we think that the name might have some effect on the model's ability to distinguish new knowledge, so we conducted experiments on KD questions for comparison. We

| Context | Alpaca | Vicuna | ChatGLM |
|---|---|---|---|
| artificial | 38.97 | 46.76 | 41.42 |
| w/ parent | 37.09 | 35.04 | 24.71 |
| w/ irrelevant | 37.12 | 37.17 | 35.00 |

Table 3: Results on KD questions with different knowledge in context.

assign two different names to artificial entities with the same properties: one is a randomly generated sequence of characters (random), and the other is a random substitution of one character (similar) for the name of its parent entity. The results of the experiments on the KD questions are shown in Table 2. We can find that the influence of names on model cognition does exist, and similar names are more likely to cause confusion in the model. However, the effect is not very large and both results are lower, which shows that the modeling of new entities by LLM is both derived from the names and influenced by the properties.

## 5.3 Impact of Provided Knowledge

To further explore the performance of LLMs when new knowledge is provided in the context, we vary the knowledge content provided for the analysis experiments.

**Parent Entity Aggravates Confusion** According to Section 5.2, the models suffer from a certain degree of confusion when faced with new knowledge that overlaps in name or properties with internal knowledge. To further verify this, we explicitly introduce parent entities in the context. Specifically, we conducted two comparison experiments: 1) adding parent entity knowledge to the context; 2) adding a random existing entity knowledge as control variables. As shown in Table 3, the performance is affected by both parent and irrelevant

| Context | Alpaca | Vicuna | ChatGLM |
|---|---|---|---|
| artificial | 27.63 | 30.1 | 27.47 |
| w/ chain | 29.21 | 33.25 | 44.43 |
| w/ irrelevant | 25.38 | 22.98 | 18.56 |

Table 4: Results on KA questions with different knowledge in context.

| | Alpaca | | Vicuna | | ChatGLM | |
|---|---|---|---|---|---|---|
| | JSON | NL | JSON | NL | JSON | NL |
| **KU** | 33.8 | 30.63 | 41.22 | 28.04 | 47.97 | 20.07 |
| **KD** | 38.97 | 36.00 | 46.76 | 36.12 | 41.42 | 22.44 |
| **KA** | 27.63 | 26.07 | 30.10 | 25.48 | 27.47 | 9.82 |
| **Avg.** | 35.09 | 32.16 | 42.74 | 31.92 | 42.56 | 20.54 |

Table 5: Results on ALCUNA with knowledge in JSON and natural language format (NL).

entities in the context, which is consistent with previous work (Shi et al., 2023). More significantly, for Vicuna and ChatGLM models, the parent entity brings more substantial performance degradation compared to the irrelevant entity, again confirming the confusion problem of existing large models in face of new knowledge.

**Chain Entities are Key to Knowledge Association** To more clearly analyze why all models performs poorly on the knowledge association problem, we conduct two additional sets of experiments on the KA questions: 1) adding knowledge about the entities involved in the reasoning chain to the context. 2) randomly sampling the same number of entities to the context for a fair comparison. The final results are shown in Table 4. We can find that the score of all models improves very much after adding the information of the entities required for inference, and the performance of all models decreases after adding irrelevant entities. This also shows that the main problem is that LLMs really cannot make the association between new and existing knowledge well, and not just the problem of not being able to make reasoning.

**Structured Knowledge is Better** Since our knowledge is represented structurally, the input of knowledge in our experiments is also structured, as shown in Table 12. To explore the effect of the form of the knowledge representation, we additionally do comparison experiments with knowledge in natural language form as input (NL). We use templates to transform each attribute into a language description for input similar to the template-based question generation process in Section 3.3. As can

be seen from Table 5, all models generally perform better with the structured input setting (JSON). The models' understanding of this structured text may come from the code in the training data. This indicates that for this kind of high-density knowledge input, a clear and structured representation is more helpful for the model's understanding.

# 6 Assess New Models with ALCUNA

In this section, we discuss how to apply the proposed ALCUNA benchmark to other models. There are two different application scenarios of our benchmark. First, if one wants to assess the knowledge understanding performance of different LLMs in the face of new knowledge, ALCUNA can be directly utilized for evaluation. On the other hand, if one wants to compare the knowledge differentiation and association abilities, the different background knowledge inside the different models may lead to an unfair comparison. Therefore, we need to conduct an additional filtering on ALCUNA to ensure the existing entities are possessed by all models, which will cause a shrinkage of our benchmark. Despite this, the current benchmark has been filtered on models such as Alpaca. A reasonable assumption is that the models that come after that will be more and more powerful, so the resulting shrinkage won't be very severe.

# 7 Related Work

**Large Language Models** In recent years, significant advancements in Large Language Models (LLMs) like FLAN-T5(Chung et al., 2022), GPT-3(Brown et al., 2020), OPT(Zhang et al., 2022), LLama(Touvron et al., 2023) and GPT-4(OpenAI, 2023) have led to exceptional performance in natural language processing tasks. At the same time, open-source LLMs based on LLama and other fundamental models for further instruction fine-tuning have emerged recently, such as Alpaca(Taori et al., 2023), Vicuna(Chiang et al., 2023), Koala(Geng et al., 2023), ChatGLM(Du et al., 2022), etc., which have also shown strong capabilities.

These models have shown breakthroughs on a variety of tasks, and some have been applied as a commercial product in daily work. Since the world is constantly changing, the ability of the models to perform when faced with new knowledge is critical.

**Existing Benchmarks** Benchmarks, such as SQuAD(Rajpurkar et al., 2016), SNLI(Bowman

et al., 2015), GLUE(Wang et al., 2018), Super-GLUE (Wang et al., 2020), LAMBADA (Paperno et al., 2016), etc., is essential for setting evaluation criteria and tracking model performance. While some traditional benchmarks like SQuAD and SNLI assess single-task abilities, GLUE and SuperGLUE evaluate general language models across various NLP tasks. More recently, with rapid development of LLMs, new benchmarks have been proposed to evaluate on more complex tasks such as college entrance exams, law school admission tests and so on (Hendrycks et al., 2021; Guo et al., 2023; Zhong et al., 2023). However, these benchmarks are still based on existing knowledge, which is abundant in the training data of LLM.

Some knowledge benchmarks (Onoe et al., 2021; Mallen et al., 2023; Arodi et al., 2023) evaluate model's ability of knowledge integration while they either only evaluate on small models which are very different with LLMs (training data, ability, etc.) or simply use knowledge that already exists. Therefore, benchmarks that assess LLMs' ability with new knowledge rather than existing knowledge are urgently needed.

**Source of New Knowledge** Many works use temporal knowledge as a source of new knowledge to evaluate new knowledge behavior of LLMs (Lazaridou et al., 2021; Agarwal and Nenkova, 2022; Jang et al., 2023; Zaporojets et al., 2023). There is also some work that uses entity or attribute substitution to create new knowledge (Longpre et al., 2022; Zhou et al., 2023). A discussion of the differences and strengths and weaknesses of our work versus prior work is in Appendix I.

## 8 Conclusion

We propose a new approach KnowGen to construct new knowledge and build an artificial biological entity benchmark ALCUNA for evaluating the ability of LLMs faced with new knowledge. We test and analyze several popular models with commonly used methods, and find some useful conclusions. Our proposed approach and benchmark can help the development of more powerful LLMs that can understand, differentiate and reason across new and existing knowledge. In the future, we expect that more LLMs will be evaluated on our benchmark. More benchmarks in other disciplines also can be constructed based our method.

## Limitations

Although the method we design can be used for the construction of any knowledge that satisfies the ontological representation, we have implemented it only on biological data. It is absolutely possible to use this approach to create new knowledge in other domains for evaluating LLMs. It is because KnowGen method for creating new knowledge only requires some defined classes, and some entities in the classes, entities with their own attributes, and connections between the entities. Any knowledge base with such a structure can be used to create new knowledge with our KnowGen method. In the future, we hope that new knowledge datasets from more domains will be available.

We evaluate only a few powerful models due to the fact that some models are closed source or the number of parameters is too large. We expect that more models can be measured using our benchmark.

## Ethics Statement

The knowledge of biological entities in our benchmark is artificially constructed, so it does not exist in the real world. Therefore, for humans, it is necessary to be careful not to be misled by the knowledge inside; for models, the hazard of using our dataset for training on model knowledge is unknown. This is in line with our expectation that we want ALCUNA to be used as a new knowledge benchmark only for the evaluation of model capabilities in face of new knowledge.

## Acknowledgements

This work was supported by National Key R&D Program of China (2021YFF0901502), National Science Foundation of China (No. 62161160339), State Key Laboratory of Media Convergence Production Technology and Systems and Key Laboratory of Science, Technology and Standard in Press Industry (Key Laboratory of Intelligent Press Media Technology). We appreciate the anonymous reviewers for their helpful comments. Xiaojun Wan is the corresponding author.

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

## A KnowGen Method

Here, we will describe knowgen in the form of natural language and math formulas.

Following the symbolic representation in Section 2.2, we divide the property triplets $\mathcal{T}(e) = \mathcal{T}_A(e) \cup \mathcal{T}_R(e)$ of parent entity into three sets: remain, change and delete, denoted as $\mathcal{T}^*(e) = \mathcal{T}_A^*(e) \cup \mathcal{T}_R^*(e)$, $* \in \{r, c, d\}$. We randomly sample some triplets from sibling entities as add set $\mathcal{T}^a(\tilde{e}) = \mathcal{T}_A^a(\tilde{e}) \cup \mathcal{T}_R^a(\tilde{e})$. Subsequently, we fuse the above four sets[5] to form the properties of artificial entity. We retain $T^r(e)$ for high similarity between the parent and artificial entities.

$$\mathcal{T}_A^r(\tilde{e}) = \{(\tilde{e}, a, v) | (e, a, v) \in \mathcal{T}_A^r(e)\}$$
$$\mathcal{T}_R^r(\tilde{e}) = \{(\tilde{e}, r, e') | (e, r, e') \in \mathcal{T}_R^r(e)\}$$

We modify the object of triplets in $T^c(e)$ with some perturbation to create uniqueness in the properties of the artificial entity, but without introducing overly unusual values.

$$\mathcal{T}_A^c(\tilde{e}) = \{(\tilde{e}, a, \text{perturb}(v)) | (e, a, v) \in \mathcal{T}_A^c(e)\}$$
$$\mathcal{T}_R^c(\tilde{e}) = \{(\tilde{e}, r, \text{perturb}(e')) | (e, r, e') \in \mathcal{T}_R^c(e)\}$$

, in which $\text{perturb}()$ represents a perturbation function that can modify values according to specific needs. For numeric attribute value $v$, we add a gaussian noise to it. For non-numeric attribute value $v$, we use the object of the same attribute from siblings entities. For association object entity $e'$, we randomly choose one of its siblings as the modified value.

We also include $T^a(\tilde{e})$ to ensure the commonality of entities within the specific class $C$.

$$\mathcal{T}_A^a(\tilde{e}) = \text{RandomSample}(\{(\tilde{e}, r, v) |$$
$$(e_i, r, v) \in \mathcal{T}_A(e_i), e_i \in Sib(\tilde{e})\}$$
$$\mathcal{T}_R^a(\tilde{e}) = \text{RandomSample}(\{(\tilde{e}, r, e') |$$
$$(e_i, r, e') \in \mathcal{T}_R(e_i), e_i \in Sib(\tilde{e})\}$$

## B Dataset Details

### B.1 Question Types Distribution

As mentioned in Section 2.4, we divide the AL-CUNA into three subsets. Each subset contains a certain number of multiple choice, fill-in-the-blank,

---

[5]We process these four collections in a manner that ensures no overlapping or shared properties between them, thereby eliminating any potential contradictions.

|  | multiple choice | Boolean | fill-in-the-blank |
|---|---|---|---|
| KU | 2459 | 4232 | 4625 |
| KD | 3487 | 34497 | 19698 |
| KA | 15353 | 0 | 0 |
| Total | 21299 | 38729 | 24323 |

Table 6: Number of different forms of KU, KD and KA questions.

| | Total | | | |
|---|---|---|---|---|
| | ChatGPT | Vicuna | Alpaca | ChatGLM |
| Refuse | 20.01 | 17.3 | 20.69 | 17.45 |
| Multi | 0.5 | 0.2 | 0.04 | 0.06 |
| Wrong | 79.49 | 82.5 | 79.27 | 82.49 |
| Total | 100.00 | 100.00 | 100.00 | 100.00 |

Table 7: The percentage of categories of incorrect responses from large models.

and Boolean questions. The specific distribution of question types is shown in Table 6. Note that we generate only multiple choice questions for the DA question set. This is done for two reasons: one is that the mutli-hop questions are too hard for current LLMs as shown in Table 1, and the other is that there may be multiple answers to one multi-hop question, and to evaluate the results more accurately and avoid false negatives, we utilize choices to limit the answer space.

### B.2 Natural Language Question Generation

We prompt ChatGPT to generate question templates given a property triplet in one-hop setting or a chain of property triplets in multi-hop setting. We provide the prompt to generate Boolean, fill-in-the-blank question templates under both settings in Figure 4, 5 and 6. We also show some generated question templates in Table 8. To generate multiple choice questions, we simply append four choices to the corresponding fill-in-the-blank questions.

In order to guarantee the accuracy of the questions generated through this approach, we select a random sample of 100 questions for both one-hop and multi-hop questions to be checked by humans. The results indicate that over 98% of the one-hop questions and over 95% of the multi-hop questions generated are accurate.

## C Introduction to Our Model We Select

We choose four representative and popular LLMs. ChatGPT, is one of the most powerful models, but is close-source, so we can only call it through the API. The other three are all excellent open-source

| Relation Chain or Property | Generated Templates |
|---|---|
| prey on | What are the animals that [T] prey on? |
| | What animals are preyed on by [T]? |
| cingulum location | Where is the cingulum located in the [T]'s mouth? |
| | What is the type of cingulum in the teeth of [T]? |
| frost free days | How many frost free days are required for the growth of [T]? |
| | How many frost-free days does the habitat of [T] have on average? |
| (have host, co-roost with) | What is the species that co-roosts with the host of [T]? |
| | What species shares a roosting habitat with the host of [T]? |
| (parasitize, visit flowers of, eat) | What is the food source of a species that feeds on the flowers visited by an organism parasitized by [T]? |
| | What is the food source of the species whose flowers are visited by an organism parasitized by '[T]'? |

Table 8: Question templates generated from properties of entities or chains of reasoning.

**System:** You are a powerful multi-hop question generator with biological knowledge. Users will provide a chain of RDF triples, and you will help write a question to ask the tail entity from the head entity. You shouldn't include bridge entities in the generated question. The question should only include the head entity [T], which is placeholder. If you can't create a valid question, reply with \"[None]\" only. Don't reply with any explanation or other information.

/* exemplars */
**User:** Given RDF triples ([T], compete with, x1), (x1, are eaten by, x2), (x2, are host of, x3), write a question to ask x3. Don't mention x1, x2, ... Write the possible question in natural English.
**Agent:** Multi-hop question: Which entities serve as the habitats for the creatures that feed on the organisms competing with [T]?
**User:** Given RDF triples ([T], parasitize, x1), (x1, prey on, x2), write a question to ask x2. Don't mention x1, x2, ... Write the possible question in natural English.
**Agent:** Multi-hop question: What is the prey of an organism that is parasitized by [T]?
**User:** Given RDF triples ([T], eat, x1), (x1, compete with, x2), (x2, are pathogens of, x3), write a question to ask x3. Don't mention x1, x2, ... Write the possible question in natural English.
**Agent:** Multi-hop question: Which species has pathogens that compete with species that are eaten by [T]?

/* query */
**User:** Given RDF triples {$Triple chain}, write a question to ask {$Tail entity}. Don't mention x1, x2, ... Write the possible question in natural English.

Figure 4: Prompt for generating multi-hop question templates.

**System:** You are a powerful question generation model with biological knowledge. Given a biological taxon's property name and a set of possible values of the property. You need to generate a yes-no question template with placeholders [T] and [V] about the given property satisfying the following constraints:
1. Most important: the generated question template must contain [T] and [V].
2. The placeholder [T] can be replaced with any taxons name.
3. The placeholder [V] can be replaced with possible value of the given property. You can refer to the given value set for the format of possible property value. But don't use any value in the value set directly.
4. Don't do the replacement, just provide the question template with [T] and [V] placeholders.
5. Try to create the question even if the property is not a biological property.
6. Don't reply with any explanation or other information.

/* exemplars */
**User:** Property name: skeleton contains   Value set: ['chitin']
**Agent:** Question template: Does the skeleton of [T] contain [V]?
**User:** Property name: body shape   Value set: ['cylinder', 'elongated', 'bilaterally symmetric', 'radially symmetric']
**Agent:** Question template: What's the body shape of [T]
**User:** Property name: litters per year   Value set: ['1.0 /year', '3.0 /year', '1.5 /year', '2.0 /year', '2.5 /year']
**Agent:** Question template: Does one [T] produce [V] litters per year?

/* query */
**User:** Property name: {$Property name}   Value set: {$Possible values of Property}

Figure 5: Prompt for generating Boolean question templates.

```
System: You are a powerful question generation model with biological knowledge. Given a biological taxon's
property name. You need to generate a question template with a placeholder [T] about the given property so that
the placeholder [T] can be replaced with any taxons name. Try to create the question even if the property is not a
biological property. Don't reply with any explanation or other information.

/* exemplars */
User: Property name: skeleton contains
Agent: Question template: Which organic compound is a component of a [T]'s skeleton?
User: Property name: body shape
Agent: Question template: What's the body shape of [T]
User: Property name: litters per year
Agent: Question template: How many litters can [T] have per year?

/* query */
User: Property name: {$Property name}
```

Figure 6: Prompt for generating fill-in-the-blank question templates.

models. Due to the limitation of GPU memory space, we choose those open-source models ranging from 6B-13B.

- ChatGPT: ChatGPT is a sibling model to InstructGPT (Ouyang et al., 2022), which is trained on a vast instruction dataset and further tuned by reinforcement learning with human feedbacks (RLHF).

- Alpaca-7B: Alpaca is an open-source instruction-following LLM trained for academic purposes, which is fine-tuned from the LLaMA 7B model (Touvron et al., 2023) on 52K instruction-following demonstrations.

- Vicuna-13B: Vicuna is trained by fine-tuning LLaMA using 70K conversations with ChatGPT shared by users. A preliminary evaluation using GPT-4 as a judge shows that Vicuna-13B achieves more than 90% of the quality of ChatGPT.

- ChatGLM-6B: ChatGLM is also an open-source instruction-following LLM, which is based on General Language Model (Du et al., 2022) framework.

We download the above three open source models from Huggingface[6], and thanks to the convenient design of the FastChat[7] library, we unify the testing framework for all models and call them through the API.

We also consider several models with 7B parameters for evaluation to compare the performance of models of same size, which may help to analyze the

impact of different training processes on the ability in face of new knowledge. We select Llama2-Chat-7B, Alpaca-7B, Vicuna-7B, and ChatGLM-7B specifically. The results are presented on Table 9.

## D Introduction to Our Experiment Method

### D.1 Zero/Few-shot Evaluation Setting

The zero-shot setting is where the model is given explicit instructions to directly complete the mission. This scenario evaluates the ability of the original model to solve the problem autonomously without training. In our benchmark, the input to the zero-shot is the new knowledge of the artificial entity and a question to be asked about it.

Compared to the zero-shot setting, in the few-shot setting the model is given several additional examples from the same task as a reference. This allows evaluating the ability of the model to learn the task quickly based on a limited number of samples, and is also consistent with practical situations where supervised training is not convenient. According to the Min et al. (2022)'s study, in our benchmark assessment, we provided 3 to 5 examples for each type of problem, expecting that it would be sufficient to be able to demonstrate the labeling space for this type of problem.

### D.2 Chain-of-Thought (CoT) Form

For both the zero-shot and few-shot evaluation settings, we add the design of the CoT form. For the zero-shot setting, we added the words "Let's think step by step." at the end of the question, expecting the model to output the thinking process, which can help LLM to reason about complex problems. For

---

[6] https://huggingface.co/
[7] https://github.com/lm-sys/FastChat

| | Llama2-Chat-7B | Alpaca-7B | Vicuna-7B | ChatGLM-6B | Llama2-Chat-7B | Alpaca-7B | Vicuna-7B | ChatGLM-6B |
|---|---|---|---|---|---|---|---|---|
| | Zero-Shot-Vanilla | | | | Zero-Shot-CoT | | | |
| **KU** | 28.92 | 31.02 | 28.00 | 34.64 | 47.75 | 39.81 | 32.74 | 24.85 |
| **KD** | 32.61 | 15.35 | 34.70 | 32.29 | 37.48 | 14.29 | 33.61 | 22.84 |
| **KA** | 18.71 | 24.60 | 23.40 | 10.29 | 24.61 | 19.66 | 17.56 | 4.97 |
| **Avg.** | 29.57 | 25.12 | 30.88 | 31.26 | 39.50 | 29.44 | 31.67 | 22.04 |
| | Few-Shot-Vanilla | | | | Few-Shot-CoT | | | |
| **KU** | 37.06 | 33.80 | 32.78 | 47.97 | 65.20 | 40.77 | 38.50 | 40.91 |
| **KD** | 44.38 | 38.97 | 39.39 | 41.42 | 44.71 | 36.24 | 38.71 | 37.19 |
| **KA** | 24.42 | 27.63 | 24.39 | 27.47 | 28.72 | 25.73 | 27.11 | 26.93 |
| **Avg.** | 39.27 | 35.09 | 35.14 | 42.56 | 50.15 | 38.02 | 37.21 | 37.64 |

Table 9: Performance of LLMs of 7B parameters at our benchmark under different settings.

| | multiple choice | Boolean | fill-in-the-blank |
|---|---|---|---|
| KU | 2025 | 3138 | 3814 |
| KD | 2486 | 24592 | 12601 |
| KA | 8757 | 0 | 0 |
| Total | 13268 | 27730 | 16415 |

Table 10: Number of different forms of KU, KD and KA questions after filtering.

the few-shot setting, we add the thought process in the answer to each sample question shown to inspire the model.

## E  Details of Our Approach to Filtering Questions

Specifically, we retain only those artificial entities whose parent entities could be perfectly recalled by the model. In addition, since answering multi-hop questions requires the model to make use of each single-hop knowledge, we then filter out any reasoning chains that contain knowledge that cannot be correctly recalled by the model.

The method used for the above two filtering is to construct question templates for the knowledge involved, including attributes and relationships, based on previous work (Petroni et al., 2019), and then to query the model using the few-shot setting. We filter samples in our benchmark for every evaluated model to ensure that our questions are specific to the ability about new knowledge, and then select the intersection of filtered questions for fair experimentation and analysis. The number of questions per category left is shown in Table 10.

## F  Analysis about Models' Output

An example of the output of the model is shown in Table 11. To better analyze the models' responses, as shown in Table 7, we divide all the models' error outputs into 3 categories, including rejecting responses, answering multiple options, and other incorrect responses.

We can find that the percentage of answering multiple options for all models is very small, which indicates that all models can understand and comply with our requirements very well. In addition, some of the questions are rejected by some models, probably because some models recognizes that it cannot answer the corresponding question and responds with "*I don't know*" or "*I am sorry*".

## G  Impact of Different Modifications

As shown in Section 5.2, models commonly struggle with knowledge differentiation when the artificial entity and the parent entity are similar. In this section, we further conduct an ablation study to investigate the specific impact of different modifications (i.e. variation and dropout).

To ablate one type of modification, we reconstruct artificial entities in the KD dataset. For each question in the original KD dataset, we have a parent entity $e^p$ and the corresponding attribute $a$ to be queried. We then reconstruct several artificial entities by modifying one attribute of $e^p$ (except $a$) at a time. From this, we obtain several artificial entities with different similarity caused by the same type of modification. For a fair comparison, we only experiment with $e^p$ with 10 attributes. We randomly sample 1000 parent entities and create a total of $1000 \times 10 = 10000$ new artificial entities. Finally, we conduct experiments by querying models about them with the same question about $a$.

The experiment results are presented in Table 7. Different models exhibit almost the same trend under both modifications. It is clear that dropout yields a stable improvement as the similarity decreases while the impact of variation is relatively weak and insignificant.

| |
|---|
| *Question*: What organism is parasitized by an organism that feeds on Agabus ajax? |
| *Answer*: Rangifer tarandus |
| *Model's Output*: "Thought process: Agabus ajax is a species of aquatic beetle that feeds on Acari, Culicidae, Daphnia magna, Daphnia pulex, and Ceratopogonidae. To answer the question, we need to identify an organism that is parasitized by an organism that feeds on Agabus ajax. Final answer: Thala gorii"] |
| *Question*: Is Sorghum versinatum a type of hyperhydate? |
| *Answer*: Yes |
| *Model's Output*: "ANSWER: Yes" |
| *Question*: What's the preferred water depth range for Plana? |
| *Answer*: 500.0 cm |
| *Model's Output*: ANSWER: 500 cm\n\nIt is important to note that the information |

Table 11: Example of question and answer from our models.

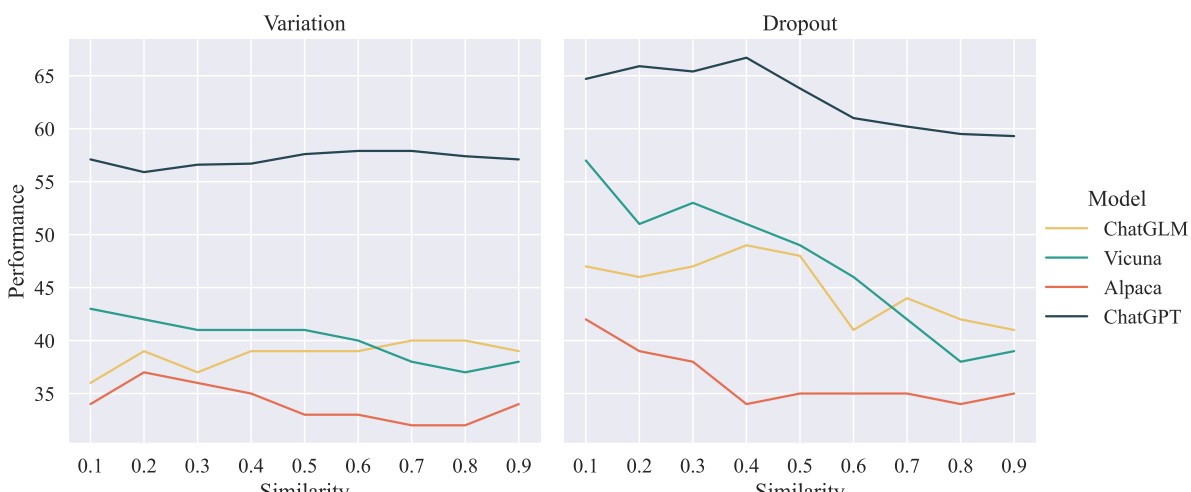

Figure 7: Relationship between model performance on KD questions and the property similarity between the artificial entity and its parental entity due to different levels of modification, i.e. variation and dropout.

## H  Prompts in Experiments

In this section, we will show prompts used in our settings. Since our questions contain Boolean, multiple choice, and fill-in-the-blank forms, the rest of the prompt is identical except for the restrictions on the output format for each type of question that are targeted. Therefore, without loss of generality, we provide the prompts for the fill-in-the-blank questions as a demonstration. We present the CoT zero-shot prompt as an example in Table 12. The other methods of prompt are basically similar.

## I  Source of New Knowledge

**Temporal Knowledge**   Using temporal knowledge as a source of new knowledge has the following disadvantages:

1. the expense of collecting data. With the continuous emergence of new LLMs, the tempo-

ral knowledge used needs to be re-collected each time, requiring labor and resources to race with the training of the LLM.

2. uncertain validity and risk of information leakage. Some LLMs do not announce the training data they use, so it is not known whether the temporal knowledge collected each time is still valid or not.

3. fairness of comparison. Since the range of timestamps for training data of each model is different, the new temporal knowledge is different for each model. Therefore, the test data used for each evaluation needs to be different, and it is uncertain whether such a comparison is fair.

In contrast, our proposed KnowGen method solves the above problems. Since it is artificial knowledge, it will be valid for a long time and no

You are a powerful question-answering system with knowledge in the field of biology.

Users will provide some biological information along with a question.

Your task is to combine the information provided by the user with your biological knowledge to answer the question.

If you are unable to answer the question, simply respond with "I don't know."

Here is the basic information about a taxon you can refer:

###

```
{
    "name": "Bainvillevillea spinosa",
    "property": {
        "cellularity": ["multicellular"],
        "conservation status": ["least concern"],
        "geographic distribution": ["Ecuador"],
        "habitat": ["terrestrial"],
        "leaf complexity": ["compound"],
        "leaf morphology": ["broad"],
        "leaf sheddability": ["evergreen"],
        "plant growth form": ["branched"],
        "produces": ["oxygen"],
        "woodiness": ["woody"]
    },
    "rank": "species"
}
```

###

Answer the following question a few words: What is the habitat of Bainvillevillea spinosa?

Desired format: Thought process: <Thought process>, Final answer: [Final answer].

Let's think step by step.

Table 12: Demonstration of the zero-shot prompt in the CoT form.

more effort is required to collect data repeatedly. Moreover, for all models, the knowledge is definitely new, so we can also use the same test data to evaluate the models, which also ensures the validity and fairness of the evaluation.

**Previous methods of entity and attribute substitutions**  As for the relationship to previous work (Longpre et al., 2022; Zhou et al., 2023), all of us "construct" knowledge, but our knowledge forms, construction purposes, and implementation methods are different.

1. Knowledge form: The knowledge in Longpre et al. (2022) and Zhou et al. (2023) is a statement of a few sentences describing a simple fact. Whereas our knowledge is represented in ontological form, with complete properties, relations and classes, which is more structured and complete.

2. Purpose of construction: both of them are designed to construct knowledge that contradicts the internal knowledge of the model, as a way to make the model hallucinatory or unreliable predictions. Instead, our work is intended to simulate the generation of new knowledge that is associated and consistent with the existing knowledge.

3. Implementation: the methods of them are limited by the form of knowledge representation, and merely construct counterfactuals by replacing the name of the entity in the sentence with the name of another existing entity. Our method, on the other hand, will create a completely new entity with a new name and which reasonably exists in the original knowledge system through operations such as heredity, variation and dropout.