# OpenReview forum: "ALCUNA: Large Language Models Meet New Knowledge"
_EMNLP/2023/Conference — EMNLP 2023 Main_

### Official Review · Reviewer_Hv6T · 2023-08-02

**Soundness:** 3

**Excitement:**

3: Ambivalent: It has merits (e.g., it reports state-of-the-art results, the idea is nice), but there are key weaknesses (e.g., it describes incremental work), and it can significantly benefit from another round of revision. However, I won't object to accepting it if my co-reviewers champion it.

**Paper Topic And Main Contributions:**

This paper aims to assess the capabilities of LLMs on knowledge understanding, knowledge differentiation, and knowledge association over new artificial knowledge. To do this, the authors introduce KNOWGEN, a knowledge generation method designed to formulate new artificial knowledge utilizing an existing ontology. KNOWGEN leverages heredity, variation, extension and dropout of knowledge triplets to construct new entities. These new entities are then used to examine the LLMs' capabilities over new knowledge. Their results reveal that the performance of current LLMs in face of new knowledge is not satisfactory.

**Questions For The Authors:**

1. Can I get a detailed example of the model's input concerning a specific query? Which related knowledge triplets are presented to the models along with the queries? Is additional knowledge about existing entities also provided?
2. Could you clarify what the KD and KA scores imply? In my view, if a model exhibits insufficient performance on knowledge differentiation, it might be due to inadequate understanding of the new knowledge or confusion brought on by existing knowledge about familiar entities. If a model performs poorly on KA, it might be because of the model's limited ability to execute multi-step reasoning or a lack of memorization of the single-hop relations within the multi-step chain.

**Reasons To Accept:**

1. Testing LLM on new knowledge is an important topic, and this paper proposes an effective dataset construction method to facilitate this.
2. The demonstration figure 1 is concise and straightforward.

**Reasons To Reject:**

1. The evaluation scope is relatively limited, with just 4 LLMs being tested. These models vary in size (Alpaca is 13B, Vicuna is 7B), resulting in an insufficiently holistic outcome and potentially unfair comparisons.
2. The authors do not provide any complete example from the dataset, and the dataset files are absent from the supplementary materials. Thus, the precise nature of the input/output for the LLMs remains vague. (Refer to question 1)
3. The scores on KD, KA are coupled with multiple capabilities, including comprehension of new knowledge, memorization of existing knowledge, and knowledge reasoning. It is hard to interpret from the current scores and reach conclusions regarding the model's capabilities. (Refer to question 2)

**Reproducibility:**

3: Could reproduce the results with some difficulty. The settings of parameters are underspecified or subjectively determined; the training/evaluation data are not widely available.

**Reviewer Confidence:**

4: Quite sure. I tried to check the important points carefully. It's unlikely, though conceivable, that I missed something that should affect my ratings.

---

> ### Author Rebuttal · Authors · 2023-08-29
>
> Thanks for your careful and valuable comments. We will explain your concerns point by point.
>
> > 1. The evaluation scope is relatively limited, with just 4 LLMs being tested. These models vary in size (Alpaca is 13B, Vicuna is 7B), resulting in an insufficiently holistic outcome and potentially unfair comparisons. (Reject 1)
>
> **A1**: Thank you for your advice. Considering the computational cost as well as the length of the paper, we evaluate several popular LLMs (Alpaca-7B, Vicuna-13B, ChatGLM-6B and ChatGPT) and focus on detailed analysis experiments.
> In fact, it is very hard for model evaluation work to evaluate a large number of large models nowadays due to computational cost and other reasons[1]. For example, AGIEval[2] evaluates only 3 models and Open-QA[3] evaluates only 4 models.
>
> We will later release our dataset to make it easy for the community to evaluate other models using our benchmarks. As more and more LLMs continue to emerge, a comprehensive evaluation of current models requires a joint effort by the entire community.
>
> In addition, following your suggestion, we have supplemented our experiments on models of the same size.
> Unlike the previous small models such as BERT, it is difficult to freely choose models of the same size for evaluation because the large models are now published by different organizations and each model is far more expensive to train than before (Alpaca, for example, does not have an official 13B version). Based on these considerations, we select Vicuna-7B[4] as well as Llama2-Chat-7B[5] (an impressive model that has recently released) for evaluation as supplementary experiments.
> The results are shown in the following table.
>
> |      | Vicuna-7B         | Vicuna-7B     | Vicuna-7B        | Vicuna-7B    | Llama2-7B-Chat    | Llama2-7B-Chat | Llama2-7B-Chat   | Llama2-7B-Chat |
> | ---- | ----------------- | ------------- | ---------------- | ------------ | ----------------- | -------------- | ---------------- | -------------- |
> |      | Zero-Shot-Vanilla | Zero-Shot-CoT | Few-Shot-Vanilla | Few-Shot-CoT | Zero-Shot-Vanilla | Zero-Shot-CoT  | Few-Shot-Vanilla | Few-Shot-CoT   |
> | KU   | 28.00             | 32.74         | 32.78            | 38.50        | 28.92             | 47.75          | 37.06            | 65.20          |
> | KD   | 34.70             | 33.61         | 39.39            | 38.71        | 32.61             | 37.48          | 44.38            | 44.71          |
> | KA   | 23.40             | 17.56         | 24.39            | 27.11        | 18.71             | 24.61          | 24.42            | 28.72          |
> | Avg. | 30.88             | 31.67         | 35.14            | 37.21        | 29.57             | 39.50          | 39.27            | 50.15          |
>
> Combined with Table 1 in the paper, we can see that Vicuna-7B generally performs worse than the Vicuna-13B, but performs better than the Alpaca-7B most of the time, and performs similarly to the ChatGLM-6B.
> And the Llama2-Chat-7B performs better than the Vicuna-7B. The most impressive thing is the much improved results under Few-Shot-CoT. This suggests that Llama2-Chat has a stronger reasoning ability, which may be due to its higher quality instruction data. The above findings are also consistent with the technical report on Llama2[5], which states that it performs the best among the open source models on reasoning tasks.
>
> **With more and more large models emerging, model evaluation is a huge undertaking. Our contribution in this paper is in designing methods to create new knowledge and in implementing the ALCUNA benchmark**. We call for a joint community effort to evaluate the ability of large models to meet new knowledge on our benchmarks, to form a well-developed leaderboard.
>
>
> > 2. Can I get a detailed example of the model's input concerning a specific query? Which related knowledge triplets are presented to the models along with the queries? Is additional knowledge about existing entities also provided? (Question 1; Reject 2)
>
> **A2**: Sorry for the inconvenience caused to you. **In fact, we have provided examples of the dataset as well as examples of the inputs to the model in the paper**.
> As mentioned in Section 5.2, "an example of prompt used in our experiment is shown in Table 11" (in line 477-478).
> And we also metion it in Appendix G: "We present the CoT zero-shot prompt as an example in Table 11. The other methods of prompt are similar" (in line 1016-1018).
> **An example of our dataset is also shown in Table 11, from which it can be seen that we do not provide existing knowledge as well as related knowledge.**
>
> The reason we do not provide them is because of the filtering we have performed in Section 5.4, which ensures that all the existing knowledge involved in our questions can be perfectly recalled by models. Therefore, there is no need to provide them and it puts the focus of our evaluation on new knowledge.
>
> Moreover, **we have also performed the comparison experiments that provide parent entities and related reasoning knowledge**, see Section 6.3. Through these experiments we have come to some conclusions: 1) Parent Entity Aggravates Confusion; 2) Chain Entities are Key to Knowledge Association; 3) Structured Knowledge is Better.
>
> Our full dataset will be released in the future.
> If you want more examples now, here is another example of input of our models with the setting few-shot prompt (to make it easier to read, we use "..." to denote excessive properties and ):
>
> ```json
> "instruction": """You are a powerful question-answering system with knowledge in the field of biology. Users will provide some biological information along with a question. Your task is to combine the information provided by the user with your knowledge of biology to answer the question. If you are unable to answer the question, simply respond with "I don't know."
> """,
> "exemplars": [
>       ["""Example 0: Based on you encyclopedic biological knowledge, answer the following question a few words: Is Ecuador part of the geographic distribution of Bainvillevillea spinosa? Reply with: "Yes" or "No".
>   Here is the basic information about a taxon you can refer:
>   ###
>   {
>     "name": "Bainvillevillea spinosa",
>     "property": {
>       "geographic distribution": ["Ecuador"],
>       "habitat": ["terrestrial"],
>       "leaf complexity": ["compound"],
>       "leaf morphology": ["broad"],
>       "leaf sheddability": ["evergreen"],
>       ...
>       "produces": ["oxygen"],
>       "woodiness": ["woody"]
>     },
>     "rank": "species"
>   }
>   ###
>         "ANSWER: Yes"
>   """
>    ],
>       ["""Example 1: Based on you encyclopedic biological knowledge, answer the following question a few words: Is Philendron xanatii found in the geographic distribution of Panama? Reply with: "Yes" or "No".
>   Here is the basic information about a taxon you can refer:
>   ###
>   {
>     "name": "Philendron xanatii",
>     "property": {
>       "geographic distribution": ["Ecuador"],
>       "plant growth form": ["siphonous"],
>       ...
>       "produces": ["oxygen"],
>       "trophic guild": ["photoautotroph"]
>     },
>     "rank": "species"
>   }
>   ###
>         "ANSWER: No"
>   """
>   ],
>       ["""Example 2: Based on you encyclopedic biological knowledge, answer the following question a few words: Is Burma part of the geographic distribution of Curculigoigochioides? Reply with: "Yes" or "No".
>   Here is the basic information about a taxon you can refer:
>   ###
>   {
>     "name": "Curculigoigochioides",
>     "property": {
>       "cellularity": ["aggregative multicellular"],
>       ...
>       "produces": ["oxygen"],
>       "trophic guild": ["photoautotroph"]
>     },
>     "rank": "species"
>   }
>   ###
>         "ANSWER: I don't know"
>   """
>   ]
> ],
> "query": """Based on you encyclopedic biological knowledge, answer the following question a few words: Does Eos squamata visit Casuarina grandis? Reply with: "Yes" or "No".
>   Here is the basic information about a taxon you can refer:
>   ###
>   {
>     "name": "Eos squamata",
>     "property": {
>       "Body symmetry": ["bilaterally symmetric"],
>       "ecoregion": ["Seram Rain Forests"],
>       "life span": ["15.2 years"],
>       ...
>       "visit": ["Casuarina grandis"],
>       "visual system": ["corneal eyes"]
>     },
>     "rank": "species"
>   }
>   ###
>   "ANSWER:"
>   """
> ```
>
>
> > 3. Could you clarify what the KD and KA scores imply? In my view, if a model exhibits insufficient performance on knowledge differentiation, it might be due to inadequate understanding of the new knowledge or confusion brought on by existing knowledge about familiar entities. If a model performs poorly on KA, it might be because of the model's limited ability to execute multi-step reasoning or a lack of memorization of the single-hop relations within the multi-step chain. (Question2; Reject 3)
>
> **A3**: Thank you for the opportunity to clarify this point. First, it is important to note that **KU, KD, and KA are the three capabilities that we define and that we design tasks to assess**. As described in lines 104-112 of the paper, when the LLM is faced with new knowledge, KU refers to the model's ability to understand the new knowledge, KD refers to the model's ability to distinguish between new and existing knowledge, and KA refers to the model's ability to associate the new and existing knowledge. These three capabilities are well understood and self-consistent.
>
> Second, if it is really desirable to split these abilities, then it is important to note that in our experiments, as described in Section 5.4 and Appendix E, **we have performed data filtering**. This means that the **model can perfectly memorize existing knowledge** now. So **both KD and KA are abilities concerning new knowledge**, i.e., KD performs poorly because of confusion between new and existing knowledge, and KA performs poorly because of the model's poor reasoning ability on new and existing knowledge.
> In addition, the experiments in Figure 3, Table 3 and Table 4 may answer your confusion. Figure 3 illustrates that the more similar the new and existing knowledge are, the worse the performance on KD. This suggests that the poor performance on KD is due to **confusion**. Table 3 illustrates that when existing knowledge is provided in context, the model performs worse on KD, which also suggests that existing knowledge causes confusion. Table 4 illustrates that the model's performance on KA improves only to a limited extent when single-hop relations in reasoning chains are provided in the context, which suggests that **the model is mainly limited by the poor ability to combine new and existing knowledge for reasoning**.
>
>
> [1] Chang, Yupeng, et al. "A survey on evaluation of large language models." arXiv preprint arXiv:2307.03109 (2023).
>
> [2] Zhong, Wanjun, et al. "Agieval: A human-centric benchmark for evaluating foundation models." arXiv preprint arXiv:2304.06364 (2023).
>
> [3] Wang, Cunxiang, et al. "Evaluating open question answering evaluation." arXiv preprint arXiv:2305.12421 (2023).
>
> [4] Chiang, Wei-Lin, et al. "Vicuna: An Open-Source Chatbot Impressing GPT-4 with 90\%* ChatGPT Quality"
>
> [5] Touvron, Hugo, et al. "Llama 2: Open foundation and fine-tuned chat models." arXiv preprint arXiv:2307.09288 (2023).

---

### Official Review · Reviewer_NN51 · 2023-08-03

**Soundness:** 4

**Excitement:**

4: Strong: This paper deepens the understanding of some phenomenon or lowers the barriers to an existing research direction.

**Missing References:**

Lazaridou, Angeliki, et al. "Mind the gap: Assessing temporal generalization in neural language models." *NeurIPS 2021.*

Zaporojets, Klim, et al. "TempEL: Linking dynamically evolving and newly emerging entities." *NeurIPS 2022.*

Jang, Joel, et al. "TemporalWiki: A lifelong benchmark for training and evaluating ever-evolving language models." EMNLP 2022.

Agarwal, Oshin, and Ani Nenkova. "Temporal effects on pre-trained models for language processing tasks." *TACL 2022.*

**Paper Topic And Main Contributions:**

This paper describes a method to generate structured datasets which can be used to evaluate the ability of LLMs on unseen (at training time) knowledge.  The authors start from the Encyclopedia of Life (EOL) - an existing structured knowledge base consisting of attributes for and relations between different biological entities. They then propose a mechanism to construct hybrid novel entities with attributes and relations derived from a single parent and multiple siblings, all belonging to the same class.  They then provide a structured representation of this entity as input to an LLM, and evaluate it on question answering for this entities on single-hop or multi-hop reasoning questions. The questions used are based on templates derived from ChatGPT. The evaluation results are provided across 4 LLMs, mainly providing evidence of characteristics of entities that influence exact-match QA performance across zero-shot, few-shot and chain-of-thought setups.

**Contributions:**

- A benchmark and method to test LLMs for ability to reason about unseen knowledge.
- Error analysis indicating the properties of novel entities and answering strategies that maximize the QA exact answer performance.

**Questions For The Authors:**

**Question A**: Why is a hybrid entity based evaluation benchmark (ALCUNA) preferable over evaluation of new entities from a temporal KBs (such as wikipedia based examples mentioned earlier).

**Question B**: Can the hybrid parent + sibling mechanism account for completely new knowledge relating to new classes being discovered? (eg. a new family/species)

**Question C**: Why is the few-shot CoT performance of models for the knowledge association task (Table 1) significantly worse? Excluding this row, few-shot CoT numbers  seems to always improve performance.

**Reasons To Accept:**

1.  A straightforward mechanism to create synthetic, algorithmically generated benchmark could help address the issue of evaluation for LLMs. This mechanism depends upon a pre-built knowledge base, and could potentially be applied to other domains (beyond the biological species).
2. The error analysis centered around entity overlap, chain entities, and the role of structured representation provides a good example of mechanisms to check the robustness of central claims of the performance of LLMs on new knowledge.

**Reasons To Reject:**

1. A lack of sufficient coverage of related work, especially on the effects of new temporal behavior of LMs (Lazaridou et al;, 2021; Zaporojects et al., 2022; Jang et al., 2022; Agarwal and Nenkova, 2022). In particular, Jang et al (2022) generate Wikipedia diffs over time and these could be an alternative mechanism for evaluating new knowledge in LLMs (assuming an appropriate prompting setup).

2.  Unclear generalizability of method for generating new entity/relations. While the EOL biological dataset provides a combinatorially large space for new entity generation, it is restricted to a specific domain. If is unclear if an ALCUNA like method would work for alternative KBs. The authors claim in the limitations that structured data is only available in the biological domain, but this is not the case (WordNet/ConceptNet/Wikidata etc).

3. Lack of baseline EOL entity QA performance: An important evaluation measure missing from the experiments is the performance of the chosen models on entities that already exist within EOL. Having these numbers would help contextualize the drop in performance for new entities.

**Reproducibility:**

3: Could reproduce the results with some difficulty. The settings of parameters are underspecified or subjectively determined; the training/evaluation data are not widely available.

**Reviewer Confidence:**

4: Quite sure. I tried to check the important points carefully. It's unlikely, though conceivable, that I missed something that should affect my ratings.

**Typos Grammar Style And Presentation Improvements:**

line 216: We → we

line 271 “mimic human progressive cognitive process of entities” a verb is missing here?

---

> ### Author Rebuttal · Authors · 2023-08-29
>
> Thanks for your careful and valuable comments. We will explain your concerns point by point.
> > 1. Why is a hybrid entity based evaluation benchmark (ALCUNA) preferable over evaluation of new entities from a temporal KBs (such as wikipedia based examples mentioned earlier). (Question A)
>
> **A1**: This is a good question, thank you for raising it. We also mention the use of temporal knowledge (i.e., using the most updated knowledge after the model has been trained) to evaluate the model in lines 83-88 of the paper. However, there are the following drawbacks:
> 1) **the expense of collecting data**. With the continuous emergence of new LLMs, the temporal knowledge used needs to be re-collected each time, requiring labor and resources to race with the training of the LLM.
> 2) **uncertain validity and risk of information leakage**. Some LLMs do not announce the training data they use, so it is not known whether the temporal knowledge collected each time is still valid or not.
> 3) **Fairness of comparison**. Since the range of timestamps for training data of each model is different, the new temporal knowledge is different for each model. Therefore, the test data used for each evaluation needs to be different, and it is uncertain whether such a comparison is fair.
>
> In contrast, our proposed KnowGen method solves the above problems. Since it is artificial knowledge, it will be valid for a long time and no more effort is required to collect data repeatedly. Moreover, for all models, the knowledge is definitely new, so we can also use the same test data to evaluate the models, which also ensures the validity and fairness of the evaluation.
>
> Reviewer wiV6 also considers our task formulation to be important and proposed method to be valuable:
> *This is an **important problem** to study, especially in context of applying models to changing knowledge. Interestingly, their task formulation **differs from prior studies on temporal new knowledge (which are facts not available to the model during training, but are often of similar of existing entities and potentially have information leakage)**...Their main contributions is a clever method to generate new knowledge, which is connected to existing knowledge... This would help study the ability of models on new knowledge, **without performing such evaluation in real time***.
>
> > 2. Can the hybrid parent + sibling mechanism account for completely new knowledge relating to new classes being discovered? (eg. a new family/species) (Question B)
>
> **A2**: First, it needs to be clarified that our aim is to **evaluate LLMs by creating reasonable new knowledge** as mentioned in Q1, not to simulate the emergence of all possible new knowledge, which is impossible because the emergence of new knowledge is a process of human cognition and the development of things.
>
> Second, the emergence of a completely new class of knowledge is rare, and if it occurs, it is perfectly fine to use it directly to test LLMs.
> In fact, our contribution is the bulk creation of reasonably new knowledge that is meant to solve the problem of not having suitable new knowledge for evaluating big models. **The emergence of entirely new knowledge is nice, and our work does not need to cover all the generation of new knowledge.**
>
> > 3. Why is the few-shot CoT performance of models for the knowledge association task (Table 1) significantly worse? Excluding this row, few-shot CoT numbers seems to always improve performance. (Question C)
>
> **A3**: We have also observed this strange phenomenon. To further explore the reasons behind it, we sampled 100 questions for analysis. We have observed several failure patterns in the model outputs.
>
> For large-scale LLM (ChatGPT):
>
> 1) Under CoT setting, model is more inclined to output "I don't know" than under Vanilla setting (58 samples v.s. 25 samples). We find that in the corresponding rationales, ChatGPT usually comes to an intermediate conclusion of "Relevant information is not provided in the context", rather than reasoning with their internal prior knowledge.
>
>
> For small-scale LLMs (ChatGLM, Vicuna, Alpaca):
>
> 1) The rationales in model outputs under CoT setting only repeat information provided in the prompt, which only relates to the first hop reasoning. It doesn't help much with multi-hop reasoning in the KA scenario.
>
> 2) LLMs under CoT setting often output failure answer 0 in a fixed string format, *"... is not specified. Therefore, the answer is 0."*. The pattern occurs 15 times in 100 samples. This phenomenon is similar to the first failure pattern of large-scale LLMs mentioned above.
>
> We list one sample for each failure pattern in the table below.
> |Question|Answer|Vanilla Output|CoT Output|
> |---|---|---|---|
> |What plant species grows on the organisms that are eaten by Hyppa recta?\n0. Chlorocebus pygerythrus\n1. Abaria electa\n2. Sus scrofa\n3. Galium aparine|3|ANSWER: 3 (Galium aparine)|None of the organisms listed are eaten by \'Hyppa recta\'. The organisms that are eaten by \'Hyppa recta\' are \'Salvia officinalis\', \'Vaccinium\', and \'Vaccinium angustifolium\'. There is no information available to suggest that any of these plants grow on any of the organisms listed. Therefore, the ANSWER is "I don\'t know."|
> |What is the host of the organism that is preyed upon by the parasite 'Eucharisaris intermedia'?\n0. Separatista\n1. Thomisidae\n2. Anura\n3. Zale aeruginosa|1|ANSWER:1|The organism that is preyed upon by the parasite 'Eucharisaris intermedia' is 'Formica fusca' or 'Formica'. Therefore, the ANSWER is [2]|
> |What is the organism parasitized by the fourth-level host of 'Pterocarparpusinalis'?\n0. Pompilus fasciatus\n1. Quercus robur\n2. Houstonia palmeri\n3. Amelanchier|1|ANSWER:1|The organism parasitized by the fourth-level host of \'Pterocarparpusinalis\' is not specified. Therefore, the ANSWER is "0".|
>
> We think this phenomenon is interesting and worth exploring further.
>
> > 4. Unclear generalizability of method for generating new entity/relations. While the EOL biological dataset provides a combinatorially large space for new entity generation, it is restricted to a specific domain. If is unclear if an ALCUNA like method would work for alternative KBs. The authors claim in the limitations that structured data is only available in the biological domain, but this is not the case (WordNet/ConceptNet/Wikidata etc). (Reject 2)
>
> **A4**: It needs to be clarified that in the Limitations section, "we have implemented it only on biological data due to to the factor of structured datasets"(line 635-637) is trying to convey that since EOL data fits well into the ontological organization, we have only created the ALCUNA benchmark via the KnowGen method as a specific implementation.
>
> But as we said earlier, "the method we design can be used for the construction of any knowledge that satisfies the ontological representation"(line 633-635). It is absolutely possible to use this approach to create new knowledge in other domains for evaluating LLMs. **It is because KnowGen method for creating new knowledge only requires some defined classes, and some entities in the classes, entities with their own attributes, and connections between the entities**. Any knowledge base with such a structure can be used to create new knowledge with our KnowGen method. Thus our method is generalizable.
>
> For example, Wikidata is also ontological [1]. There are many items in Wikidata, which have different properties and relations.
> In addition, items have two special properties: *instance of* (P31) and *subclass of* (P279) which can be used as *Class* in our paper. So Wikidata can also be used to create new knowledge by KnowGen.
>
> > 5. An important evaluation measure missing from the experiments is the performance of the chosen models on entities that already exist within EOL. Having these numbers would help contextualize the drop in performance for new entities. (Reject 3)
>
> **A5**: It is possible that our lack of emphasis has led to your misunderstanding. We would like to clarify that "we retained only the artificial entities whose parent entities could be recalled by the models perfectly" (line 976). You can refer to Section 5.4 and Appendix E for the details about filtering. In other words, **the performance of chosen models on the filtered existing entities within EOL is one hundred percent correct**. Therefore, we eliminate the influence of unknown exisiting entities (since none of them exists) to independently analyze the impact of new knowledge, and hence "compare the models’ performance in face of new knowledge in a more focused and fair way"(line 476).
>
> > 6. Typos and missing references. (Reject 1, Typos)
>
> **A6**: Thank you for your patience and careful review, we will fix these typos.
> In addition, utilizing temporal knowledge as new knowledge for evaluating models is indeed a relevant kind of solution, so we will add these papers to the related work section in our camera-ready version.
> We apologize for our mistake and thank you again for your suggestions and corrections.
>
> [1] Wikidata: A large-scale collaborative ontological medical database

---

### Official Review · Reviewer_wiV6 · 2023-08-05

**Soundness:** 4

**Excitement:**

4: Strong: This paper deepens the understanding of some phenomenon or lowers the barriers to an existing research direction.

**Paper Topic And Main Contributions:**

The paper aims to evaluate the ability of current LLMs to reason in the presence of new knowledge (concepts that are different or new from existing corpora). For this, they first propose a method to construct new entities that are similar to existing entities in a structured knowledge but different in some properties. Using Encyclopedia of Life as the structured knowledge they construct 3k such new entities. Using prompting mechanisms, they generate questions (via templates) to probe different models. Finally they study 4 LLMs and present findings about the ability of LLMs to answer questions about new entities and new knowledge.

This is an important problem to study, especially in context of applying models to changing knowledge. Interestingly, their task formulation differs from prior studies on temporal new knowledge (which are facts not available to the model during training, but are often of similar of existing entities and potentially have information leakage). The formulation here tests models on the ability to process and generalize to completely new entities which are still grounded to existing knowledge. Their main contributions is a clever method to generate new knowledge, which is connected to existing knowledge (is similar in many ways) but is still sufficiently new. This would help study the ability of models on new knowledge, without performing such evaluation in real time. The benchmark would be useful in addition to test knowledge editing and propagation methods like [1, 2], to test how extensively models use new knowledge.

They evaluate LLMs by providing new knowledge as part of the input via prompting. Their experimental findings suggest that while ChatGPT is able to answer simple questions on new knowledge, it still struggles with more complex reasoning. All other models (smaller model sizes) struggle with simple and complex questions. The findings are interesting and were important to experimentally establish. They highlight that injecting knowledge of new entities might not be easily achievable via simple prompting or providing the knowledge as part of input.

[1] De Cao et, al 2021 - Editing Factual Knowledge in Language Models

[2] Onoe et, al 2023 - Can LMs Learn New Entities from Descriptions? Challenges in Propagating Injected Knowledge

**Questions For The Authors:**

Question A: What is the naming scheme for new entities for the overall experiments ? There is an experiment showing the effect of naming scheme, but which one is chosen for the rest of the experiments?

Question B: Line 373 - What does the “number of Google search terms” criteria used for identifying specialized entities mean?

Question C: For the experiment in Fig 3 - can you keep same entities but vary the number of properties that are inherited/varied during construction? I think this would provide a more careful analysis of the influence of similarity.

**Reasons To Accept:**

1. Clever, simple and scalable method to construct artificial benchmark of new knowledge.
2. New entities are not entirely random or non-sensical, but are connected to part of existing ontology. The entity construction algorithm takes commonalities and variations in classes into account to ensure new entities are similar in some extent to existing entities. This helps truly test the generalization to new entities of similar type.
3. Constructed new knowledge is controllable to allow varying of overlap with existing knowledge.
4. Experiments are detailed - studying the LLM’s abilities via zero and few shot prompting and by varying overlap, modifying names, etc. Experiments and findings go deeper to provide few hypotheses for the model behaviour (entity confusion)

**Reasons To Reject:**

1. While the paper has many experiments, there are a couple of experiments which I think are important to solidify the findings:

	i) Performance of models on existing knowledge - the baseline of how these models perform on reasoning about organisms in general is important establish the drop in performance with new knowledge. This would answer whether the model struggles with KA in general when queried about organisms or whether this is specific to new knowledge. If models struggle with existing knowledge, the issue could be the usage of structured knowledge from the input rather than using “new” knowledge.

	ii) Performance across modifications - Fig 3 shows performance across entity similarity. Extending such analysis to different levels of modification (heriditary v/s variation v/s dropout), ablating one modification type at a time would help understand what kinds of knowledge change models struggle with.

2. Comparability across models - as the data for evaluation for each model is different (filtering Sec 5.4) - are the results comparable across models ? Experimenting with entities new to all models might help address this concern.

If the experiments and clarifications are added to the paper, I'm willing to increase my score.

**Reproducibility:**

4: Could mostly reproduce the results, but there may be some variation because of sample variance or minor variations in their interpretation of the protocol or method.

**Reviewer Confidence:**

4: Quite sure. I tried to check the important points carefully. It's unlikely, though conceivable, that I missed something that should affect my ratings.

---

> ### Author Rebuttal · Authors · 2023-08-29
>
> Thanks for your careful and valuable comments. We will explain your concerns point by point.
> > 1. Performance of models on existing knowledge - the baseline of how these models perform on reasoning about organisms in general is important establish the drop in performance with new knowledge... (Reject 1.i)
>
> **A1**: Thank you for your thoughtful advice.
>
> First, we would like to clarify that we have already filtered all multi-hop questions to ensure that existing knowledge involved in each single-hop can be recalled by the model as described in Appendix E (line 978-982).
>
> Second, we have in fact conducted a similar experiment to explore the reasons for the lower performance on KA in Section 6.3, Chain Entities are Key to Knowledge Association (line 588-590). We found that including all problem-related knowledge (i.e. all chain entities including both artificial and existing entities) in the context leads to performance improvement in KA. To some extent, it shows that the main problem is the inability of LLMs to make the association between new knowledge (provided in context) and existing knowledge (recalled from model parameters), rather than a lack of reasoning ability.
>
> We totally agree with you about the baseline of reasoning about existing entities, which eliminates the influence from artificial entities.Here we report the results in the table below. For each multi-hop question in KA about artificial entity $\tilde e$ with triplets chain $\mathcal{T}_C = (\tilde e, r_1, e_1)\rightarrow ...  $, we utilize the parent entity $e^p$ with the same relation value $(e^p, r_1, e_1)$ to construct another triplets chain $\mathcal{T}_C' = (e^p, r_1, e_1)\rightarrow ...  $, and create a multi-hop question about existing knowledge only.
>
> |            | questions about artificial entities | questions about existing entities |
> |------------|--------|--------|
> | alpaca-7b  | 27.72  | 39.20  |
> | vicuna-13b | 29.53  | 53.13  |
> | chatglm    | 27.76  | 59.76  |
> | chatgpt    | 41.46  |  86.82  |
>
> As shown in the table, there is a large performance gap between the two experiment settings, which demonstrates that the reason models struggle with KA is due to the influence of new knowledge.
>
> > 2. Performance across modifications - Fig 3 shows performance across entity similarity. Extending such analysis to different levels of modification (heriditary v/s variation v/s dropout), ablating one modification type at a time would help understand what kinds of knowledge change models struggle with. (Reject 1.ii, Question C)
>
> **A2**: Thank you for your advice. To avoid misunderstanding, we firstly revise that questions in KD set only relates to the attributes of variation and dropout, and the similarity is in fact measured by the proportion of heredity.
>
> Here we reconstruct artificial entities in the KD dataset to analyse the influence of different modification (dropout or variation). For each question in the original KD dataset, we have a parent entity $e^p$ and the corresponding attribute $a$ to be queried. We then construct several artificial entities by modifying one attribute of $e^p$ (except $a$) at a time. From this, we obtain several artificial entities with different similarity caused by the same type of modification. For a fair comparison, we only experiment with $e^p$ with a fixed number of attributes (10 in specific). We randomly sample 1000 parent entities and create 1000 * 10 = 10000 new artificial entities. The result is shown below.
>
> ***Variation***
>
> |   similarity    | 0   | 0.1 | 0.2 | 0.3 | 0.4 | 0.5 | 0.6 | 0.7 | 0.8 | 0.9 |
> |-------|-----|-----|-----|-----|-----|-----|-----|-----|-----|-----|
> | ChatGLM |0.36|0.39|0.37|0.39|0.39|0.39|0.4|0.4|0.39|0.38|
> | Vicuna  |0.43|0.42|0.41|0.41|0.41|0.4|0.38|0.37|0.38|0.35|
> | Alpaca  | 0.34|0.37|0.36|0.35|0.33|0.33|0.32|0.32|0.34|0.36|
> | ChatGPT |0.571|0.559|0.566|0.567|0.576|0.579|0.579|0.574|0.571|0.566|
>
>
> ***Dropout***
>
> | similarity      | 0   | 0.1 | 0.2 | 0.3 | 0.4 | 0.5 | 0.6 | 0.7 | 0.8 | 0.9 |
> |-------|-----|-----|-----|-----|-----|-----|-----|-----|-----|-----|
> | ChatGLM |0.47|0.46|0.47|0.49|0.48|0.41|0.44|0.42|0.41|0.38|
> | Vicuna  |0.57|0.51|0.53|0.51|0.49|0.46|0.42|0.38|0.39|0.35|
> | Alpaca  |0.42|0.39|0.38|0.34|0.35|0.35|0.35|0.34|0.35|0.36|
> | ChatGPT | 0.647|0.659|0.654|0.667|0.638|0.61|0.602|0.595|0.593|0.564|
>
>
> Results on different models exhibit almost the same trend. It is shown that dropout yields a stable improvement as the similarity decreases while the effect of variation is relatively weak and insignificant.
>
>
> > 3. Comparability across models - as the data for evaluation for each model is different (filtering Sec 5.4) - are the results comparable across models? Experimenting with entities new to all models might help address this concern. (Reject 2)
>
> **A3**: Yes, we totally agree with you. In fact **that's exactly what we do**! A detailed description of how we filtered the data is shown in Appendix E (line 988-992):
>  "*We filter samples in our benchmark for every evaluated model to ensure that our questions are specific to the ability about new knowledge, and then select the **intersection** of filtered questions for fair experimentation and analysis*."
>
> We are very sorry for the confusion caused by our lack of clarity.
>
> >  4. What is the naming scheme for new entities for the overall experiments ? There is an experiment showing the effect of naming scheme, but which one is chosen for the rest of the experiments? (Question A)
>
> **A4**: We apologize for commenting out the description of the entity naming scheme in our experiment due to a typo.
> The reason we use SIMILAR and RANDOM naming scheme for the experiment in Table2 is that we want to reflect the effect of the name on the model more clearly through these two extreme naming schemes.
>
> In the real world, when a new creature is found, the naming scheme is usually to use the same prefix or suffix as creatures of its own species.
>
> In other experiments, **in order to mimic the real world scenario** and considering the tokenization algorithms used in LLMs, we firstly split the names of related existing entities (parent entity and sibling entities of parent entity) into **subwords** through one tokenizer (WordPiece tokenizer). Then we randomly select the name of related entities, and **for the i-th selected entity we choose its i-th subword**. For example, "ALCUNA" is created from **AL**PACA and VI**CUNA**.
>
> Sorry again for our typo. This experiment setting is important to state, and we will add it in the camera ready version. Thank you very much for your careful review of our paper.
>
> > 5. Line 373 - What does the “number of Google search terms” criteria used for identifying specialized entities mean? (Question B)
>
> **A5**: Since the EOL dataset contains almost all of the world's organisms, some of them may be quite rustic and cannot be recognized by almost all models in our preliminaries. With the same purpose as the filtering of the data in Section 5.4, we first use Google search to determine whether an entity is common or not. For these entities that we get less than 10K search results, we discard them. In this way, we obtain an initial filtered list of common entities. Then we use models to filter the entities left to ensure that these entities are known by them as described in Section 5.4.
> **In short, Google search is to get more common entities, which is a preliminary operation prior to model filtering.**

---

### Meta-Review · Area_Chair_T2EX · 2023-09-12

**Recommendation:** 4

**Metareview:**

The paper presents an interestingly clever approach for generating new knowledge connected to existing one. This is used to create a new benchmark to assess LLMs' abilities in reasoning about this knowledge and show that existing LLMs still struggle with the task by running experiments of 4 well known LLMs. Overall, all reviewers found the paper convincing and well written. Discussion was lively and constructive, and authors proved knowledgeable and willing to take into account reviewers’ suggestions for further improving their paper. The paper seems mature and timely interesting for the community.

**Pros.**

-  the paper presents a smart method to generate (artificial) new knowledge, which could be particularly useful for evaluating LLMs

- it tackle an important problem, i.e. reasoning about new knowledge;

- it shows experimentally that the task is challenging also for top LLMs;

- it proposes a new framing of a known task;

- it presents interesting error analyses.

**Cons.**

- the paper does not clearly report data about comparability of performances against existing knowledge and across models. However, details were convincingly provided during the discussion.

- some related works seems not accounted for (see reviews)

---

### Decision · Program_Chairs · 2023-10-07

**Decision:**

Accept-Main

**Comment:**

The paper presents an interestingly clever approach for generating new knowledge connected to existing one. This is used to create a new benchmark to assess LLMs' abilities in reasoning about this knowledge and show that existing LLMs still struggle with the task by running experiments of 4 well known LLMs. Overall, all reviewers found the paper convincing and well written. Discussion was lively and constructive, and authors proved knowledgeable and willing to take into account reviewers’ suggestions for further improving their paper. The paper seems mature and timely interesting for the community.

**Pros.**

-  the paper presents a smart method to generate (artificial) new knowledge, which could be particularly useful for evaluating LLMs

- it tackle an important problem, i.e. reasoning about new knowledge;

- it shows experimentally that the task is challenging also for top LLMs;

- it proposes a new framing of a known task;

- it presents interesting error analyses.

**Cons.**

- the paper does not clearly report data about comparability of performances against existing knowledge and across models. However, details were convincingly provided during the discussion.

- some related works seems not accounted for (see reviews)